# Comparative analysis of rhizosphere microbiomes of cultivated and wild rice under contrasting field water regimes

Yuhong Luo,[1] Xiaolong Xu,[1] Renfei Qiao,[1] Ru-Peng Zhao,[1] Zu-Wen Zhou,[1] Dong-Ao Li,[1] Yuhao Wen,[1] Jia-Ming Song,[2] Ling-Ling Chen[1,3]

**ABSTRACT**  Asian cultivated rice (*Oryza sativa L.*) is domesticated from the common wild rice (*Oryza rufipogon Griff.*). However, the increasing water stress caused by climate change in recent years has become a major threat to rice growth and yield. To explore the adaptive changes of rhizosphere microbiomes in annual cultivated and perennial wild rice under different water limitation conditions, we conducted metagenomic sequencing analysis on rice rhizosphere soil samples from natural environments with varying water conditions. In particular, the genus *Pseudomonas* plays a dominant role in the rhizosphere microbiome of wild rice under non-irrigated condition. For archaea, we found that, compared to non-irrigated condition, submergence condition enriched methanogenic *Methanosarcina*. In comparison to cultivated rice, wild rice showed significant enrichment of *Nitrosarchaeum*, as ammonia-oxidizing archaea play a key role in the nitrogen cycle, whereas cultivated rice tends to enrich methanogenic archaea (*Methanosarcina*), which may increase methane emissions and have adverse environmental impacts. The rhizosphere metabolites of wild rice also enriched DL-Nor-leucine, L-Phenylalanine, and Palmitic acid, which may enhance root water absorption and provide essential nutrients to help rice resist water-limiting stress. In terms of rhizosphere microbiome function, *asnB* and *nirK* were particularly enriched in wild rice under non-irrigated condition, suggesting that wild rice may exhibit higher ecological adaptability to water stress by enhancing nitrogen assimilation and denitrification processes. Excavating these microbiome communities and functional changes in rice rhizospheres is crucial for optimizing water-limiting resistance, protecting the environment, and improving rice yield.

**IMPORTANCE** This study highlights the differences in rhizosphere microbiomes and metabolites between wild and cultivated rice, providing new insights into how water limitation impacts their interaction with the rhizosphere microbiome. These findings are crucial for advancing rice cultivation under submergence and non-irrigated conditions, offering strategies to optimize farming practices, manage water scarcity, and reduce methane emissions. In contrast to cultivated rice, wild rice may regulate its rhizosphere microbial community to enhance resistance to water stress. This discovery offers valuable theoretical support for improving rice growth and adaptation across diverse ecological environments.

**KEYWORDS**  cultivated and wild rice, water stress, rhizosphere microbiome, metagenome, co-occurrence network, N cycle

**Peer Reviewer** Firouz Abbasian, University of South Australia, Adelaide, South Australia, Australia

Address correspondence to Jia-Ming Song, jmsong@swu.edu.cn, or Ling-Ling Chen, llchen@gxu.edu.cn.

The authors declare no conflict of interest.

Asian cultivated rice (*Oryza sativa L.*) is domesticated from the common wild rice (*Oryza rufipogon Griff.*) and is the main staple food for nearly two-thirds of the global population. Compared to cultivated rice, wild rice, as a perennial plant, exhibits outstanding resilience to stresses and can also be converted into biofuels and

10.1128/spectrum.00263-25  **1**

contribute to efforts in mitigating climate change, including reducing greenhouse gases and promoting soil nutrient cycling (1, 2). As the world's largest rice producer and consumer, China contributes about 26% of global rice production (https://www.fao.org). Meanwhile, global rice yields are facing increasingly severe climatic challenges—with the average expected loss rate due to drought can reach 13.1% (3). Against this backdrop, according to the China Statistical Yearbook 2023, rice production in Guangxi Province is likewise significant: its sown area is 1,760.8 thousand hectares and total output is 10.304 million tons, both ranking within the top 10 nationwide. Notably, the disaster-affected crop area in Guangxi is 219.8 thousand hectares, of which 107.2 thousand hectares were impacted by drought, accounting for 48.8%. This highlights the potential threat that localized drought poses to yield security. Previous studies have shown that rhizosphere microbiomes can improve plant growth and enhance crop adaptation to future water-limiting stresses (4, 5). However, cultivation and artificial selection have disrupted interactions between plants and beneficial microbiomes (6). The domestication of plants (7) and excessive fertilizer use (8) have led to a reduction in plant microbial diversity. In response to this phenomenon, the microbiome rewilding hypothesis proposes that rewilding modern agricultural varieties by reintroducing beneficial ancestral traits is a method to enhance the sustainability of modern agricultural systems (9).

The growth and persistence of rhizosphere bacteria in the soil are highly influenced by water stress (10). Root exudates are an important way of communication between plants and microorganisms. The indirect effects of drought can alter the impact characteristics of microbial communities involved in basic metabolic processes. Over longer timescales, changes in microbial community composition, as well as ecological evolutionary feedbacks, horizontal gene transfer, and adaptation between plants and microorganisms, can determine plant-microbe drought responses (11). Therefore, the utilization of microorganisms helps to combat climate-induced changes, with plants reorganizing their microbiomes as a "cry for help" against biotic and abiotic stresses (12). In particular, non-pathogenic microorganisms residing in the plant rhizosphere can promote plant growth, increase yield and crop quality, while also enhancing the crop's resistance to abiotic stresses.

Rhizosphere microbial communities are directly influenced by water scarcity and indirectly affected by physiological and biochemical changes in the host plant (13), particularly between wild and cultivated plants. Recent studies have shown that although the flavonoid content in artificially cultivated *Scutellaria baicalensis* is lower than that in wild *Scutellaria baicalensis*, the relative abundance of drought-resistant microorganisms in the rhizosphere of wild *Scutellaria baicalensis* significantly increases (14). In bananas, metagenomic analysis revealed genotype-driven differences in key taxa and protective functions in the rhizosphere microbiomes of wild and cultivated bananas, providing valuable data sets for future plant protection research (15). In rice, drought stress markedly reshapes the rhizosphere microbial community structure, characterized by the systematic enrichment of Actinobacteria and Firmicutes and a concomitant decline in other taxa, while drought induces persistent shifts in the endosphere microbiome. Rice actively modulates microbial recruitment by altering root exudate metabolism including increasing abscisic acid, reducing jasmonic acid and amino acids, thereby establishing a bidirectional feedback mechanism (16, 17). Genotype-specific responses further refine this process: drought-tolerant varieties preferentially enrich beneficial taxa through coordinated metabolic networks, whereas drought-sensitive cultivars exhibit more pronounced metabolic dysregulation. This plant–microbe co-adaptive mechanism provides a theoretical foundation for developing microbe-assisted drought-resistance strategies (16–19). Previous studies found that the phylogenetic distance between *Oryza* species in cultivated and wild rice showed only weak correlation (20). Therefore, difference in host genotype leads to variations in rhizosphere microbiomes, and utilizing beneficial rhizosphere microorganisms from wild plants to enhance rice's adaptation to abiotic stresses is feasible. However, our understanding of

the rhizosphere microbiome structure of wild and cultivated rice under water stress in natural environments remains limited.

Beneficial microbiomes and metabolites associated with roots and plant tissues can help alleviate plant stress through various mechanisms. Recently, the physiological and biochemical mechanisms of plant rhizosphere microbiomes in response to water stress and water use efficiency have been extensively studied (16). In general, bacterial-mediated plant drought resistance may be attributed to increased water absorption (21), changes in plant hormone levels (22, 23), alterations in antioxidant status (24), production of bacterial exopolysaccharides (EPS), accumulation of compatible solutes in plants, including amino acids, sugars, and polyamines, and increased soluble phosphorus levels in the rhizosphere (25). For example, the well-known and extensively studied *Pseudomonas* spp. effectively alleviates drought stress in *Arabidopsis* (10). Inoculation with *Pseudomonas* spp. strains containing 1-aminocyclopropane-1-carboxylate (ACC) deaminase partially mitigates the adverse effects of drought stress on pea (*Pisum sativum L.*) growth, yield, and maturity (26). Wild soybean may recruit beneficial *Pseudomonas* populations to combat salt stress by secreting key metabolites such as purines (27). The nitrogen cycling functional genes associated with these microorganisms also play important roles.

In this study, we sampled rhizosphere soil from cultivated rice in paddy fields under non-irrigated condition and submergence condition, as well as from wild rice *Oryza rufipogon* DP15 in natural ecosystems. We investigated the rhizosphere microbiomes through high-throughput sequencing and identified rhizosphere metabolites in cultivated rice and wild rice under non-irrigated condition. We found that *Pseudomonas* spp. act as important beneficial bacteria in rice to resist water stress. Additionally, we found that wild rice and cultivated rice have different preferences for archaea and exhibit environmental specificity, with cultivated rice and rice under submerged condition being more likely to enrich *Methanosarcina*. Through co-occurrence network analysis, we also found that, whether bacteria or archaea, the interactions in the rhizosphere microbiomes of wild rice and rice under non-irrigated condition were more complex. The rhizosphere metabolites of wild rice also enriched DL-Norleucine, L-Phenylalanine, and Palmitic acid, which play important roles in other abiotic stresses. This study explored the differences in rhizosphere microbiome communities due to host genotype (wild rice vs cultivated rice) and environmental conditions (non-irrigated vs submerged) and further revealed the unique microbial ecological adaptation strategies of wild rice in response to water stress, providing potential references for future rice crop improvement.

## MATERIALS AND METHODS

### Experimental design and soil sampling

Soil samples were collected from the rhizosphere of common wild rice (*Oryza rufipogon*) DP15 in the wild rice conservation plot located in Nanning, Guangxi Province, China (22°84' N, 108°28' E). DP15 is a core germplasm of common wild rice from Guangxi, provided by the State Key Laboratory for Conservation and Utilization of Subtropical Agro-bioresources. We collected cultivated rice samples from the paddy field nearest to the wild rice site, with a straight-line distance of less than 1,000 m; both the wild and cultivated rice planting areas are located at Guangxi University (Fig. S1). We selected healthy rice plants at the heading stage and collected rhizosphere soil samples from cultivated rice *Nipponbare* (OS) and wild rice DP15 (DP) under unirrigated condition on 17 May 2023. On 18 September 2023, we collected samples from the same location under irrigated conditions for *Nipponbare* (OSW) and DP15 (DPW). Loose soil around the roots was shaken off, and rhizosphere soil was collected with three replicates for each condition. A total of 12 metagenomic rhizosphere soil samples were collected across two varieties (wild rice/cultivated rice) × two conditions (nonirrigated/submerged) × three biological replicates. Samples for metabolomic analysis were taken only under the nonirrigated condition, totaling 12 (two varieties (wild rice/cultivated rice) × 6 biological

replicates). All samples were obtained from individual plants. The collected rhizosphere soil was immediately placed into sterile 50 mL centrifuge tubes, flash-cooled on-site in a portable dry-ice container, rapidly transported to the laboratory, and stored at −80℃ until subsequent DNA extraction and sequencing.

## Metagenomic microbial taxonomic classification

According to the manufacturer's instructions, total DNA from each soil sample was extracted using the MagPure Stool DNA KF Kit B (MAGEN, Guangzhou, China). Library preparation was performed using the MGIEasy Universal DNA Library Prep Set (MGI-Shenzhen, China). All soil DNA samples were sequenced on the DNBSEQ platform using a paired-end 150 bp (PE150) strategy. For each sample, an average of 17.63 Gb of raw data was generated, and the basic sequence information is listed in Table S1.

First, Fastp (v 0.20.1) (28) was used to remove low-quality metagenomic reads (length <50 bp or quality score <20). Kraken2 (v2.1.3) (29) was used for taxonomic classification of the quality-filtered data. Bracken (v 2.9) (30) was used to generate species abundance tables. Rarefaction curves were constructed with the "rarecurve" function from the vegan package (31). Principal coordinate analysis (PCoA) based on Bray-Curtis distance was performed using the R package vegan (31). Alpha diversity analysis was conducted using the same R package, vegan (31). Pairwise Adonis tests and variance partition analysis (PERMANOVA, 999 permutations) using the R package Amplicon were applied to calculate differences in β-diversity (32). Differential species abundance analysis was performed using EdgeR (33) for species with average relative abundance >0.001% in rice varieties. Species with a $P$-value < 0.05 were defined as significantly different in abundance. Diversity and relative species abundance data were visualized using the R package Amplicon (32). Bacterial and archaeal taxonomy was visualized using iToL v.7 (34). Box plots and scatter plots were visualized using the R package ggplot2 (35).

## Co-occurrence network construction

To infer the co-occurrence patterns of bacteria and archaea, the "rcorr" function from the Hmisc package (36) was used to calculate the correlation between bacteria and archaea in the top 500 samples based on their relative abundance. Only Pearson correlation coefficients with an absolute value >0.8 and statistical significance ($P$ < 0.05) were considered robust and were then submitted to the graph. adjacency function in the igraph package (37) to construct an undirected network. Network visualization and calculation of parameters, such as total nodes, total links, positive correlation, negative correlation, graph density, average clustering coefficient, and average degree, were performed using Gephi (v 0.10.1) (38) to describe the network topology.

## Extraction and identification of metabolites

Six rhizosphere soil samples each from wild and cultivated rice (biological replicates) were collected. Soil metabolites were extracted with LC–MS grade solvents (methanol A454-4, acetonitrile A998-4; Thermo Fisher Scientific, USA) containing isotope-labeled internal standards (d$_3$-Leucine, 13C$_9$-Phenylalanine, d$_5$-Tryptophan, and 13C$_3$-Progesterone). Homogenized, vortexed, centrifuged at low temperature, vacuum-dried, and reconstituted in ultrapure water–methanol for UPLC–MS analysis. Metabolites were separated and detected using a Waters ACQUITY UPLC I-Class Plus system (Waters, USA) coupled to a Thermo Scientific Q Exactive high-resolution mass spectrometer (Thermo Fisher Scientific, USA). A BEH C18 column (1.7 µm, 2.1 × 100 mm, Waters, USA) was used. In positive ion mode, the mobile phases were 0.1% formic acid in water (A) and 0.1% formic acid in methanol (B). In negative ion mode, the mobile phases were 10 mM ammonium formate in water (A) and 10 mM ammonium formate in 95% methanol (B). The gradient was 0–1 min, 2% B; 1–9 min, 2%–98% B; 9–12 min, 98% B; 12–12.1 min, 98%–2% B; 12.1–15 min, 2% B. The flow rate was 0.35 mL/min, the column temperature

45°C, and the injection volume 5 µL. MS1 and MS2 data were acquired on the Q Exactive. The scan range was $m/z$ 70–1,050; MS1 resolution 70,000, AGC target $3 \times 10^6$, and maximum injection time (IT) 100 ms. Data-dependent $MS^2$ was acquired on the top 3 precursor ions; MS2 resolution 17,500, AGC target $1 \times 10^5$, and maximum injection time 50 ms; stepped NCE was 20, 40, and 60 eV. ESI source parameters were sheath gas flow rate 40; auxiliary gas flow rate 10; spray voltage (|kV|) +3.80 in positive mode and –3.20 in negative mode; capillary temperature 320°C; auxiliary gas heater temperature 350°C.

## Data processing and statistical analysis of metabolomics

The raw data collected by LC-MS/MS were imported into Compound Discoverer (Thermo Fisher Scientific) for data processing, including peak extraction, intra- and inter-group retention time calibration, adduction combinations, missing value imputation, background peak annotation, and metabolite identification. Finally, molecular weight, retention time, peak area, and identification results were determined. Mass spectrometry data analysis was performed using the BMDB Database (BGI MDB, BGI Metabolome Database, China), mzCloud (https://www.mzcloud.org/), and ChemSpider (http://www.chemspider.com/). The data preprocessing steps were conducted using MetaX (39) as follows: (i) normalization of the data using probabilistic quotient normalization to obtain relative peak areas; (ii) correction of batch effects using Quality Control-based signal sequence (QC-RLSC); (iii) removal of compounds with a coefficient of variation in relative peak area greater than 30% in all QC samples. The identified metabolites were classified and functionally annotated using the Kyoto Encyclopedia of Genes and Genomes (KEGG) pathways (40) to understand the classification and functional characteristics of different metabolites.

Using the R package ropls (https://bioconductor.org/packages/release/bioc/html/ropls.html), Partial Least Squares Discriminant Analysis (PLS-DA) was employed to evaluate the reproducibility of metabolites within the same sample based on metabolite peak area and to reflect inter-group differences. This method uses partial least squares regression to model and predict sample categories and selects differential metabolites through the importance of variables projected (VIP). The screening criteria for differential metabolites were as follows: (i) VIP value ≥1 and (ii) $P$-value < 0.05.

## Metagenome assembly, genome binning, MAG classification, and quantification of MAGs

After independently sequencing each sample, we merged the reads from all samples and performed a metagenomic co assembly using MEGAHIT v1.2.9 (41) with default parameters and a minimum overlap of 200 bp to obtain higher quality contigs. Contigs were clustered using the binning tools in the Metawrap workflow v1.3.0 (42), including MaxBin2 (v2.2.6) (43), CONCOCT (v1.0.0) (44), and MetaBA2 (v2.12.1) (45) to reconstruct metagenome-assembled genomes (MAGs). Next, CheckM2 v1.0.1 (46) was used to assess the quality of the MAGs. After filtering, a total of 214 genome drafts were obtained, with thresholds of completeness >50% and contamination <10%. After redundancy removal with dRep (47) at 99% nucleotide identity, the final set contained 214 non-redundant MAGs. The MAGs were classified using the Genome Taxonomy Database Toolkit (GTDB-Tk v2.3.2) (48) and the GTDB Release 214 taxonomy. Then, CoverM (v 0.6.1) (49) was used to map the clean reads from each sample to the bins to obtain the final abundance for each bin. The tree was constructed using GTDB-Tk (v2.3.2) (48) and visualized with iToL v.7 (34).

## Gene prediction

Gene prediction was performed on the assembled contigs using Prodigal (v2.6.3) (50) with the -p meta option. Gene completeness was determined based on the gene predictions from Prodigal. Complete genes were retained, where complete genes are those predicted to have both a start codon and a stop codon (Prodigal indicator

"00"). A non-redundant gene catalog was constructed using CD-HIT (v4.8.1) (51). Gene abundance was calculated by mapping high-quality reads from each sample to the non-redundant gene catalog using Salmon (v1.10.2) (52).

## Functional annotation by NCycDB

The merged shotgun metagenomic sequences were searched using the nitrogen cycle gene family database, N cycling database (NCycDB) (53). NCycDB is a curated integrated database designed for fast and accurate metagenomic analysis of nitrogen cycle genes, encompassing all the latest nitrogen cycle genes (53). The entire NCycDB were used for metagenomic analysis of nitrogen cycle communities. To strike a balance between speed and accuracy, the DIAMOND (v 2.0.11) (54) program was selected to perform a nucleotide sequence search against NCycDB in blastx mode. Parameters including -k 1 -e 0.0001 were used for DIAMOND. Functional profiles were then obtained using the Perl scripts provided by NCycDB.

## Statistical analysis of data

At least three replicates were set for each experiment, and the data given here for the traits are the average value ± SE of all environments. The data were processed and analyzed using R software (v 4.2.1). For the differential abundance analysis, we added the following: data were analyzed using repeated-measures ANOVA, followed by Tukey's post hoc test ($P < 0.05$).

## RESULTS

### Overall differences in bacterial and archaeal community structures between wild and cultivated rice rhizospheres

To assess the microbial community differences between wild rice and cultivated rice, we collected rhizosphere soils from both wild rice and cultivated rice under natural conditions. The rarefaction curves reaching saturation indicate that the sequencing depth was sufficient to fully capture the rhizosphere microbial diversity in each sample (Fig. S2). In the bacterial community of rice rhizosphere soil, using Bray-Curtis dissimilarity, the submergence condition clearly clustered into one group, and significantly ($R =$ 0.443, $P = 0.003$) separated from the non-irrigated condition along the first principal coordinate (47.62%) (Fig. 1A), with different microbial community patterns between the groups (Fig. 1B). In the archaea community of rice rhizosphere soil, the cultivated rice group clearly clustered into one group, and significantly ($R = 0.392$, $P = 0.003$) separated from the wild rice group along the first principal coordinate (89.48%) (Fig. 1D). Under the four different environmental conditions in the comparative analysis, we observed no statistically significant difference in the Alpha diversity levels between cultivated rice and wild rice. However, it is worth noting that in both cultivated rice and wild rice samples, the microbial community under the non-irrigated condition exhibited higher α-diversity compared to the submergence condition (Fig. S3).

Bacteria exhibited the highest abundance in all rhizosphere samples, with 37 bacterial phylum and 1,252 bacterial genera. Among all bacterial phylum, Proteobacteria accounted for 64.94% to 68.41% of the total abundance in each sample, followed by Actinobacteria (20.09% to 24.14%) and Planctomycetes (3.82% to 40.45%) (Fig. S4). Most of the variation in bacterial microbial composition was explained by the condition. Adonis tests further revealed that both condition and the grouping of rice varieties caused changes in the bacterial community composition. Among them, the condition had the greatest impact on microbial composition, explaining 44.16% of the variation in the microbial data set, followed by host genotype (from wild rice and cultivated rice), which explained 25.85% of the variation (Table S2).

To further illustrate the soil bacterial community changes induced by water stress in different host genotypes, we identified significantly enriched/depleted bacterial species between each group based on edgeR analysis. Notably, common wild rice responded

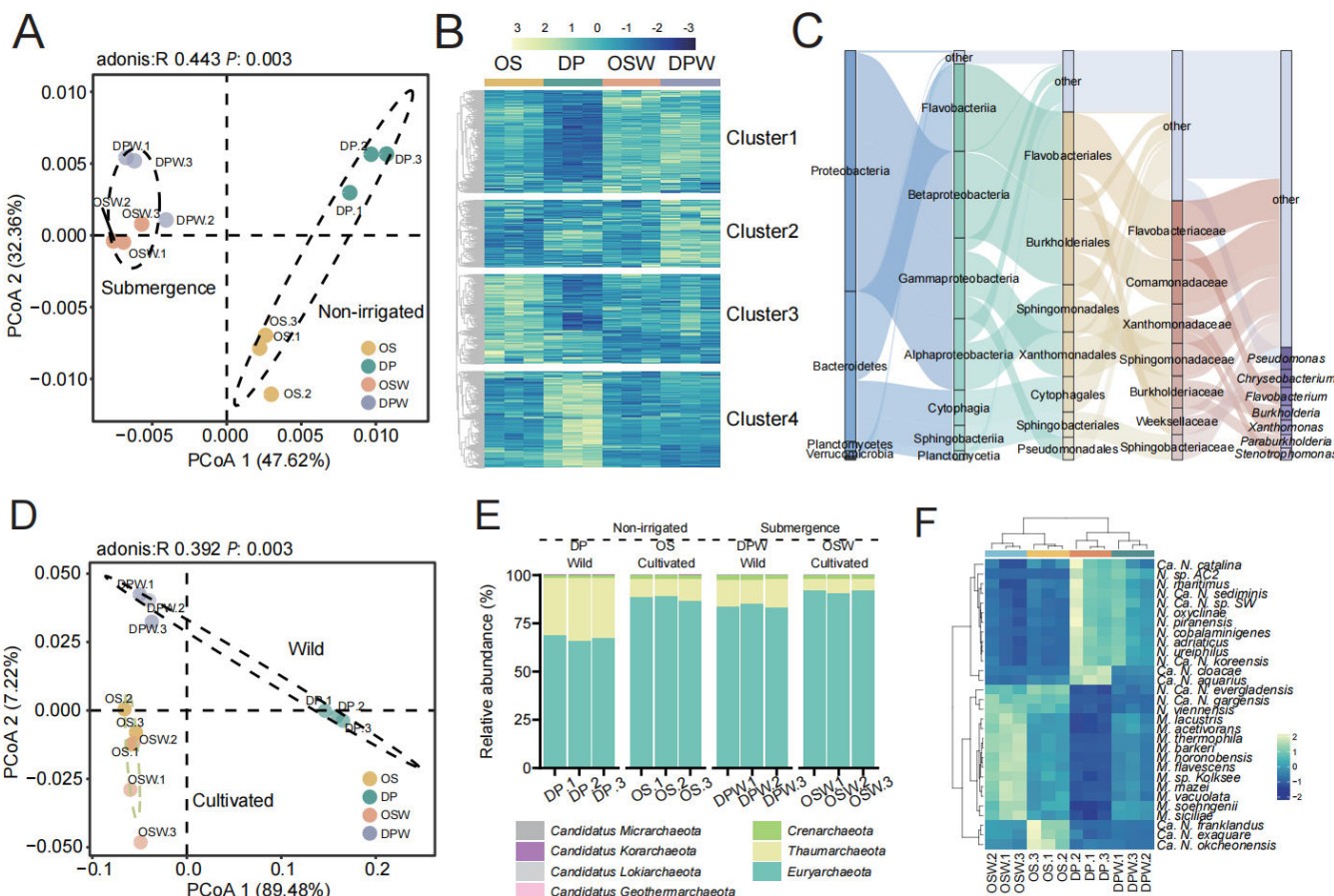

**FIG 1** Characteristics of archaeal and bacterial communities in cultivated and wild rice under submergence and non-irrigated conditions. (A) Beta diversities of bacterial communities for all samples. Colors denote the experimental groups: yellow corresponds to OS, green to DP, orange to OSW, and blue to DPW. (B) Heatmap showing the relative abundance of 4,319 bacterial taxa across different groups. The relative abundance values were scaled and displayed in varying colors, with yellow indicating high expression and blue representing low expression. (C) The Sankey diagram illustrates the taxonomic distribution (from phylum, class, order, family, to genus) of 113 differential bacteria originating from Cluster 4. These differential bacteria exhibit significant variation only under water stress conditions in wild rice and between wild and cultivated rice in non-irrigated conditions. (D) Beta diversities of archaea communities for all samples. Colors denote the experimental groups: yellow corresponds to OS, green to DP, orange to OSW, and blue to DPW. (E) The relative abundance of archaeal phylum in rice rhizosphere soil under different conditions. (F) Heatmap showing the relative abundance of differential archaea in the rhizosphere of wild and cultivated rice under both submergence and non-irrigated conditions. The relative abundance values were scaled and displayed in varying colors, with yellow indicating high expression and blue representing low expression. OS: *Oryza sativa* (cultivated rice) under non-irrigated conditions; OSW: cultivated rice under water (submergence condition); DP: wild rice DP15 under non-irrigated conditions; DPW: wild rice DP15 under water (submergence condition).

more strongly to water stress than cultivated rice, with 2,276 differentially abundant bacterial species (Fig. S5; Table S3), while only 726 species responded in cultivated rice. When comparing different host genotypes, it was found that under the non-irrigated condition, the number of differential species between varieties was 2,108. Under the submergence condition, the difference between varieties was 496 (Fig. S5). Therefore, compared to submergence conditions, host genotype had a greater impact on the microbial community changes under the non-irrigated condition. It was also found that the rhizosphere microbiota of perennial wild rice could better respond to water stress. More specifically, under the non-irrigated condition, the use of plant-beneficial *Pseudomonas* spp. was the key genus. A total of 33 bacterial species were annotated, including genera such as *Chryseobacterium* (25) and *Flavobacterium* (25), which are important for drought resistance in wild rice (Fig. 1C). Species differences mediated by host genotype between cultivated and wild rice became particularly prominent only under the specific context of water-limiting stress. Meanwhile, microbial community changes in wild rice under different water limitation conditions were also particularly drastic, reflecting not

only the severe impact of water stress on wild rice microbiota but also revealing unique strategies that wild rice may adopt when adapting to extreme environments.

Archaea in rice rhizosphere microbiota are an important source of methane production. There were 7 archaeal phylum and 119 archaeal genera. Among all archaeal phylum, Euryarchaeota accounted for 65.83% to 91.88% of the total abundance in each sample, followed by Thaumarchaeota (5.90% to 32.60%) (Fig. 1E). Most of the variation in archaeal microbial composition was explained by the variety. Adonis tests further revealed that both condition and rice varieties caused changes in archaeal community composition. Among them, host genotype had the greatest impact on microbial composition, explaining 34.14% of the variation in the microbial data set, followed by condition, which explained 33.19% of the variation (Table S4). Both wild rice and cultivated rice enriched methane-producing archaea (*Methanosarcina*) under the submergence condition compared to the non-irrigated condition (Fig. 1F). Therefore, based on our observations, when rice can maintain its normal growth and development, adopting a strategy to moderately reduce irrigation may help mitigate the environmental impact of methane-producing microorganisms. Compared to cultivated rice, wild rice exhibited significant enrichment of ammonia-oxidizing archaea (AOA) *Nitrosarchaeum* under both non-irrigated and submergence conditions (Fig. S6). This microbial community structure may be more beneficial for environmental protection because ammonia-oxidizing archaea play a key role in the nitrogen cycle, while cultivated rice tends to enrich methane-producing archaea (*Methanosarcina*), which may increase methane emissions and have adverse effects on the environment.

## Rice bacterial and archaeal rhizosphere microbial co-occurrence network patterns

To determine the co-occurrence patterns of bacteria and archaea in the rice rhizosphere, we integrated samples from wild rice and cultivated rice under submergence condition and non-irrigated condition. Using co-occurrence network analysis, we constructed co-occurrence networks for the top 500 most abundant bacteria and all archaea. The results showed that the co-occurrence patterns of bacteria and archaea in the rhizosphere soil of rice were influenced by host genotype and irrigation mode. Topological analysis revealed that the bacterial networks of cultivated, wild, non-irrigated, and submergence contained 31,072, 67,131, 56,896, and 18,395 edges, respectively (with positive correlations accounting for 74.40%, 59.86%, 58.60%, and 94.75%, respectively) (Fig. 2; Table S5). In the co-occurrence networks, compared to cultivated rice, wild rice exhibited increased topological attributes (such as total links, graph density, clustering coefficient, and average degree) (Table S5), indicating increased complexity in the bacterial and archaeal co-occurrence network of wild rice. Compared to non-irrigated condition, the topological attributes (such as total links, graph density, clustering coefficient, and average degree) of submergence condition were significantly smaller (Table S5), suggesting that under non-irrigated conditions, the complexity of the bacterial and archaeal co-occurrence network increased. Our results indicate that under non-irrigated conditions and in wild rice, rice may enhance the interactions within their rhizosphere, thereby improving adaptability.

## Variations of soil metabolomics between the cultivated and wild rice under non-irrigated conditions

Under the non-irrigated condition, the rhizosphere microbial community structure and co-occurrence network of wild rice and cultivated rice exhibited significant differences. To further explore the impact of this condition on the metabolome, we compared the metabolite profiles of wild rice and cultivated rice under the non-irrigated environment. PLS-DA analysis of 336 metabolites distinguished wild rice and cultivated rice under water-limiting conditions, indicating significant differences in the response of wild rice and cultivated rice to water-limiting conditions (Fig. 3A). The top five major metabolites in each group were alcohols, tropolones, acetylenes, and fatty acyls (Fig. 3B). Notably,

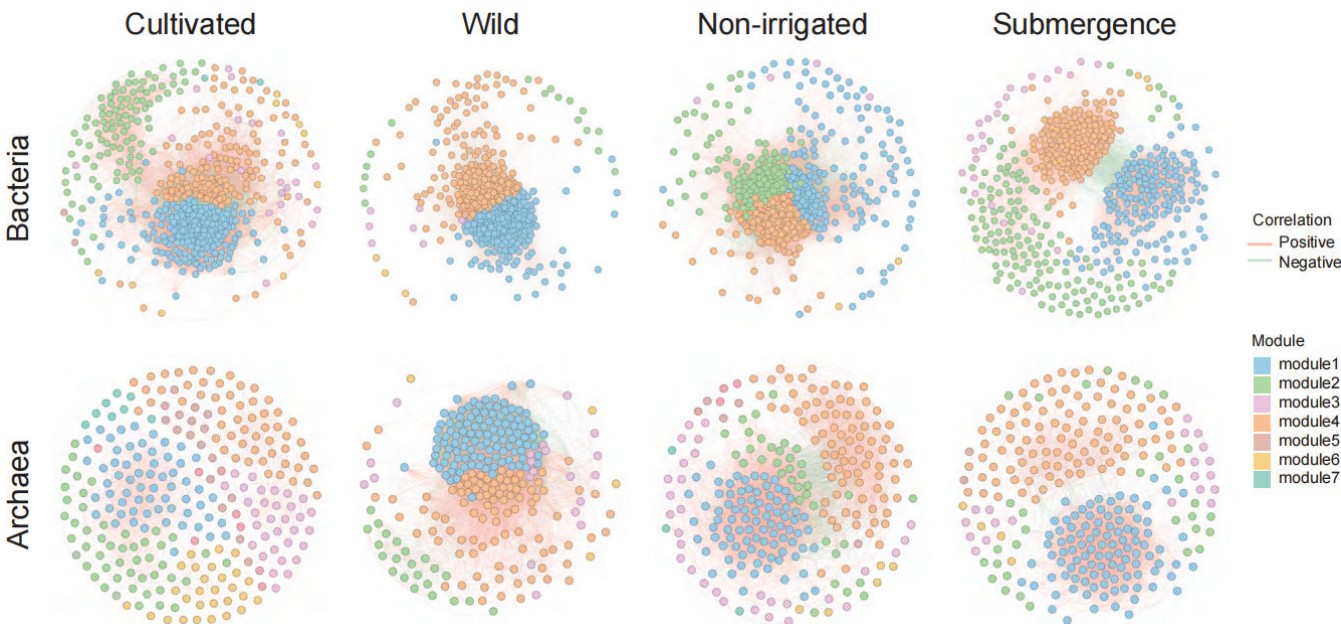

**FIG 2** Co-occurrence networks of bacteria and archaea for wild and cultivated rice in submergence and non-irrigated condition. Yellow indicates positive correlations, while green indicates negative correlations. The co-occurring networks of different compartments under different conditions. The modules of networks were presented in different colors.

the proportions of Fatty acyls in the cultivated group and wild group were 17.14% and 53.69%, respectively, while Amino acids accounted for 0.69% and 4.70% in the cultivated and wild groups, respectively (Fig. 3B).

We identified 23 differential metabolites (Fig. S7), and the rhizosphere metabolites of wild rice were enriched with DL-Norleucine, L-Phenylalanine, and Palmitic acid (Fig. 3C). Palmitic acid can significantly promote plant water-limiting tolerance through various mechanisms, including metabolic adjustments and interactions with rhizosphere microorganisms. For example, a study on *Haloxylon salicornicum* showed that palmitic acid plays a crucial role in enhancing water-limiting resistance, supporting physiological

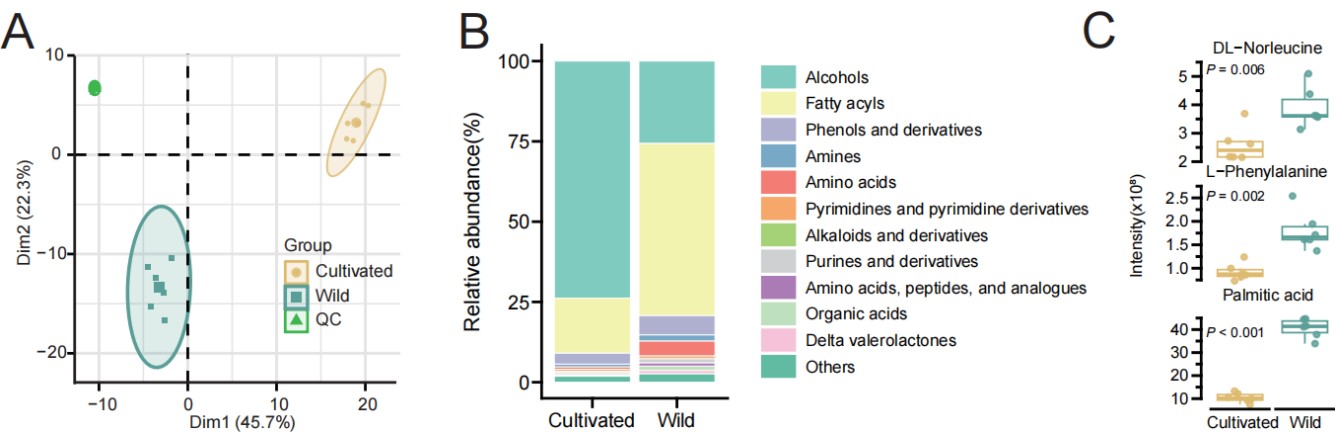

**FIG 3** Soil metabolome profiles of wild rice and cultivated rice under non-irrigated conditions. (A) Principal component analysis of the intensities of 336 metabolites in cultivated and wild rice under non-irrigated conditions. The first two principal components explain 45.7% and 22.3% of the variance, respectively. (B) The stacked bar chart illustrates the relative abundance of functionally annotated metabolite categories between wild rice and cultivated rice under non-irrigated conditions. (C) Box plots show the distribution of metabolite intensities of DL-Norleucine, L-Phenylalanine, and Palmitic acid between wild rice and cultivated rice under non-irrigated conditions. In the boxplots, the center line represents the median, the box limits indicate the upper and lower quartiles, the whiskers extend to 1.5 × the interquartile range, and the points represent outliers. Statistical analysis of these data was performed using *t*-test.

adaptations that help plants cope with water stress (55). In rhizosphere microbiome, some metabolites, indeed, have a regulatory effect on bacteria and archaea. For instance, L-Phenylalanine, as one of the root exudates, has been found to promote the growth of specific beneficial microorganisms, thereby enhancing the plant's tolerance to stress. Plants secrete various compounds from their roots, such as organic acids and amino acids, to support microbial proliferation. These metabolites can stimulate microbial activity around the roots, contributing to enhanced stress resistance (56–58).

## The MAG data set under different water stress conditions in wild and cultivated rice

To better characterize the perennial wild rice and common wild rice, we present a data set of rhizosphere metagenomes from all samples. A total of approximately 264 G of metagenomic reads were generated, with an average sequencing depth of 17.63 Gb/sample. By implementing three binning methods (MaxBin2, CONCOCT, and MetaBA2), 214 non-redundant MAGs (metagenome-assembled genomes) with more than 50% completeness and less than 10% contamination were obtained across the three rice varieties. Among this collection, 39 MAGs were classified as high-quality draft MAGs (with completeness above 90% and contamination below 5%). Figure 4A and B provide an overview of the assembly statistics for medium-quality draft MAGs, including genome size, number of bins, completeness, and contamination. The most of MAGs were assigned to bacterial phylum, including Pseudomonadota ($n = 33$), Bacteroidota ($n = 25$), and Chloroflexota ($n = 24$). Three MAGs were classified at the species level, indicating the presence of many unknown taxa in the soil.

The composition patterns of different functional features and taxa showed differences. The overall relative abundance of different functional features was relatively uniform across samples, with a few exceptions (Fig. 4C; Fig. S8). In contrast, even at the phylum level, the taxonomic composition showed significant differences. Among them, Pseudomonadota, Myxococcota, and Bacteroidota showed particularly significant abundance in wild rice under submergence conditions (Fig. 4C and D). In all MAGs recovered from rice rhizosphere soil samples, Pseudomonadota dominated in relative abundance (ranging from 21.6% to 39.7%), and it exhibited significant differences compared to other treatments (Fig. 4C and D). Within these 214 MAGs, there are many bacteria within the phylum Pseudomonadota that respond to water stress in wild rice under non-irrigated conditions.

## The gene data set under different water stress conditions in wild and cultivated rice

To characterize the functional potential of the rice rhizosphere soil microbiome, we established a gene catalog of rice rhizosphere soil microbiomes containing 29,239,492 non-redundant genes. The gene lengths ranged from 60 bp to 30,210 bp, with a median of 513 bp, of which 5,718,984/29,239,492 were complete open reading frames (Fig. 5A). Based on the gene functional annotation results from NCycDB (Nitrogen Cycling Database), the relative abundance distribution of nitrogen cycling-related genes was analyzed under different conditions (non-irrigated vs submerged) and between wild rice and cultivated rice. It was found that the abundance of different genes varied between samples (Fig. 5B). Under non-irrigated conditions, the relative abundance of *asnB*, which is involved in nitrogen assimilation, and *nirK*, which encodes the reductase for the denitrification process (nitrite reductase), was significantly higher in wild rice compared to cultivated rice ($P < 0.05$) (Fig. 5C). This suggests that wild rice may exhibit higher ecological adaptability to water-limiting stress by enhancing nitrogen assimilation and denitrification processes. Additionally, the abundance distribution of other key nitrogen cycling genes (such as *glnA, nmo,* and *gs_K00265*) was relatively stable between cultivated and wild rice (Fig. 5C). However, the overall analysis showed that wild rice exhibited higher diversity of nitrogen cycling-related genes under dryland conditions, particularly functional genes related to nitrogen assimilation and denitrification. These results

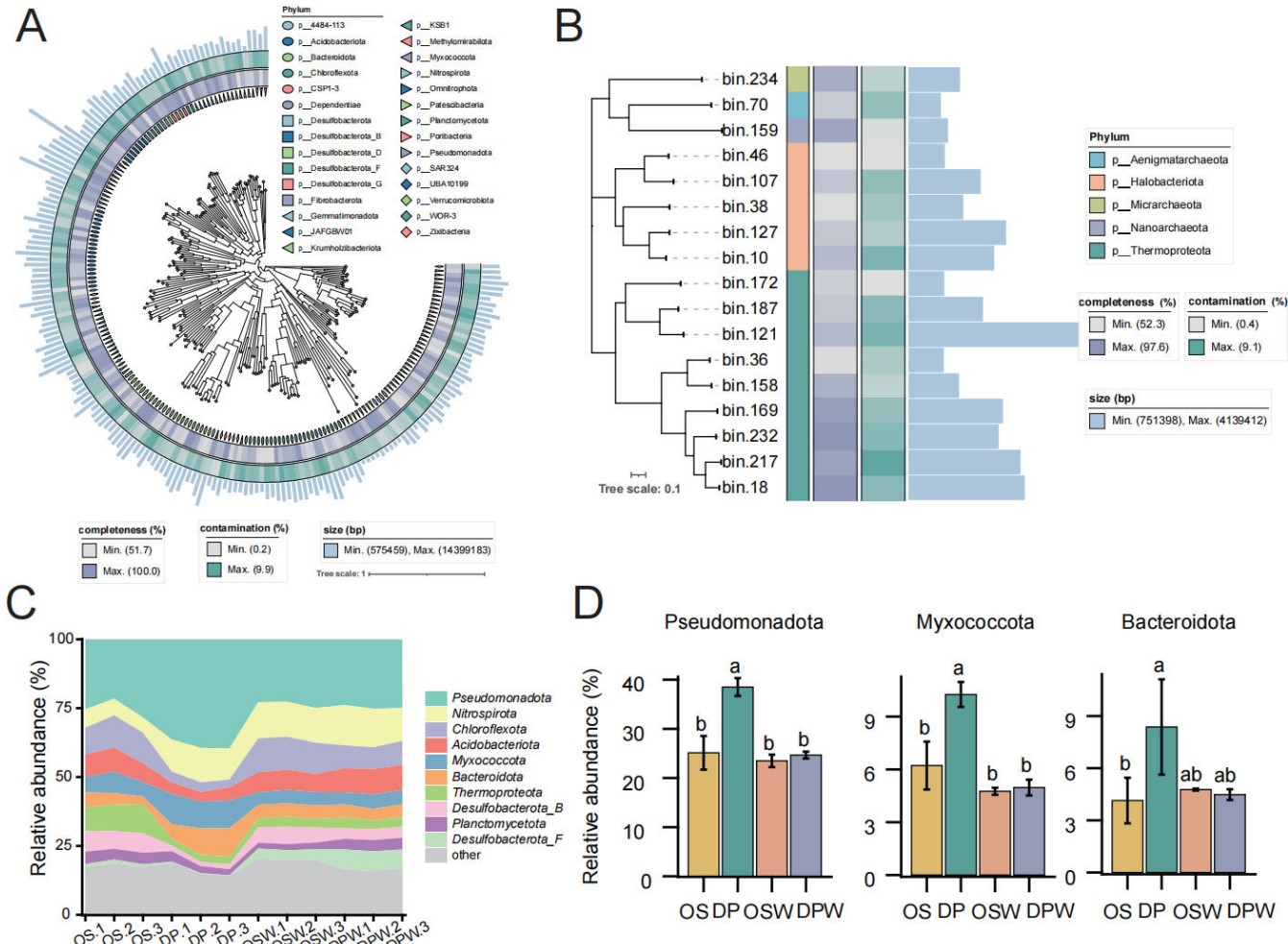

**FIG 4** Phylogenetic analysis and abundance of MAGs in rice rhizosphere soil under different conditions. Phylogenetic tree of MAGs. (A) The left figure represents bacterial MAGs, while (B) the right figure represents archaeal MAGs. In each figure, the rings from inside to outside indicate phylum, completeness, contamination, and genome size. (C) The relative abundance of microbial phylum in all samples. (D) Relative abundance of MAGs from three bacterial phylum (Proteobacteria, Firmicutes, and Actinobacteria) in rice rhizosphere soil under different conditions. Letters represent significantly different post hoc pairwise comparisons via Tukey's test ($P < 0.05$, $n = 3$). OS: *Oryza sativa* (cultivated rice) under non-irrigated conditions; OSW: cultivated rice under water (submergence condition); DP: wild rice DP15 under non-irrigated conditions; DPW: wild rice DP15 under water (submergence condition).

suggest that wild rice may have higher nitrogen utilization efficiency and environmental adaptability, especially in water-limited ecological environments.

## DISCUSSION

The soil harbors a rich, unknown microbial community, often referred to as "microbial dark matter," which constitutes a treasure trove of unexplored microbial diversity and genetic information (59). Considering the strong resilience of perennial wild rice under stress conditions, especially in water-limiting environments (60, 61), this study selected cultivated rice and wild rice as research subjects. We characterized the changes in the rhizosphere microbiome of wild rice and cultivated rice under non-irrigated and submergence conditions. Under non-irrigated conditions, we explored the microbes involved in wild rice's water stress response. Furthermore, we found that cultivated and wild rice under submergence condition might be more likely to produce methane (Fig. 6).

Under water-limiting stress, wild rice may specifically enrich certain bacterial groups and enhance their positive interactions within the roots to improve adaptability (62). We

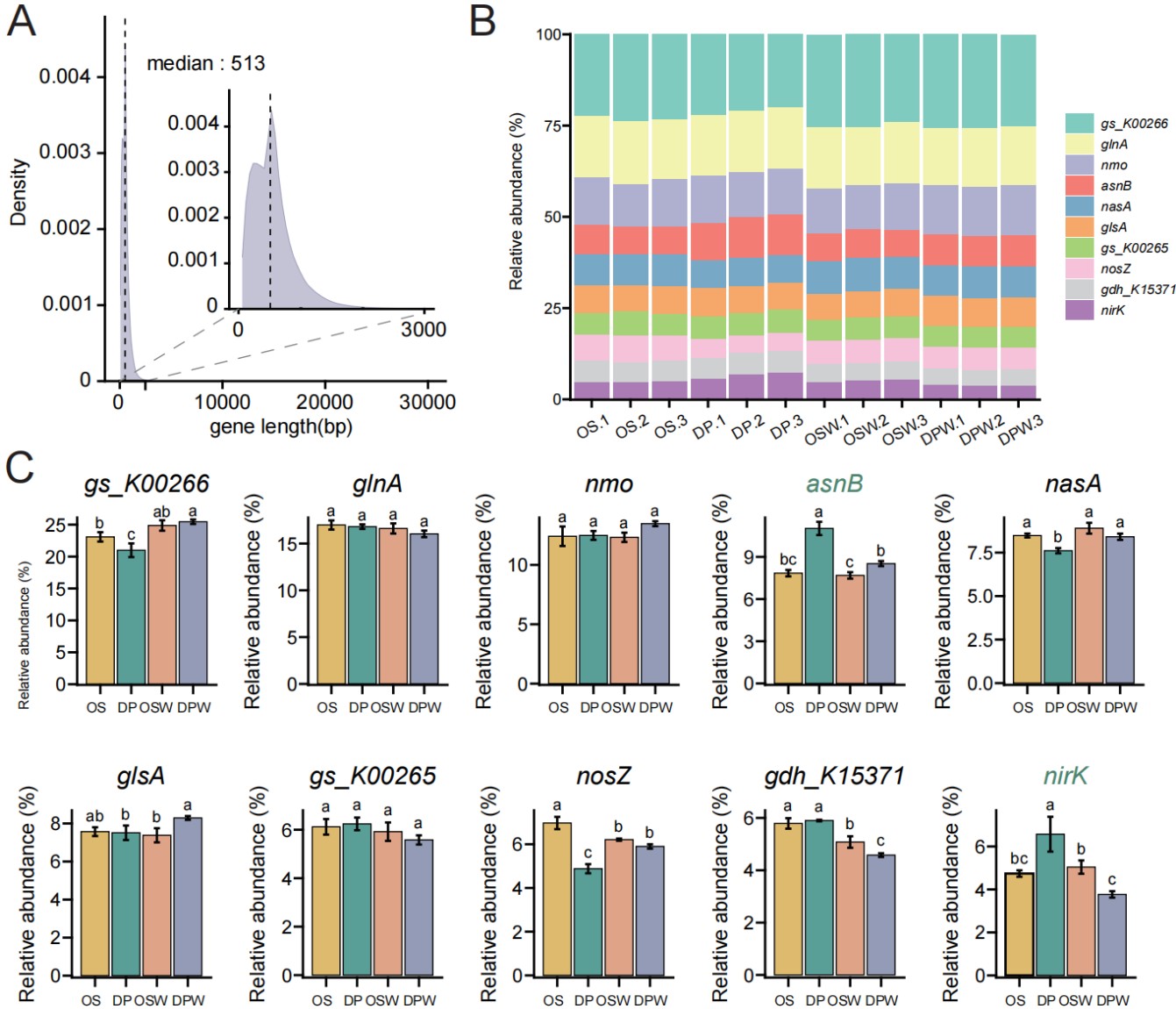

**FIG 5** Comparative gene assemblies and functional gene abundance in the rhizospheres of wild and cultivated rice under different conditions. (A) Gene length density of rhizosphere microbiomes in rice rhizosphere soil. The plot zooms in on the 0–3 kb range. The dotted line indicates the median transcript length. (B) The relative abundance of nitrogen cycle genes in all samples. Only representative gene families with high relative abundances were annotated in the figure. (C) Relative abundance of gene families with high relative abundances in rice rhizosphere soil under different conditions. Letters represent significantly different post hoc pairwise comparisons via Tukey's test ($P < 0.05$, $n = 3$). OS: *Oryza sativa* (cultivated rice) under non-irrigated conditions; OSW: cultivated rice under water (submergence condition); DP: wild rice DP15 under non-irrigated conditions; DPW: wild rice DP15 under water (submergence condition).

found that, compared to cultivated rice, *Pseudomonas* were enriched in the rhizosphere soil of wild rice. We found that, compared to cultivated rice, the rhizosphere microbial community of wild rice exhibited more complex microbial interactions, consistent with previous studies (20). Notably, beneficial microbial genera, such as *Pseudomonas* spp., *Chryseobacterium* (e.g., *Chryseobacterium indologenes*), and *Flavobacterium*, were enriched. Recent research has shown that *Pseudomonas*, a group of highly conserved microbes, is significantly enriched in plant roots and rhizosphere microbial communities, enhancing the plant's resistance to abiotic stresses such as salt stress (27) and water-limiting stress (63). *Chryseobacterium* bacteria assist in plant remediation and promote plant growth (64). *Flavobacterium* directly promotes and regulates plant growth by facilitating nutrient cycling and providing beneficial plant hormones (65–67) and grows well in water-limiting and salt-tolerant environments (68).

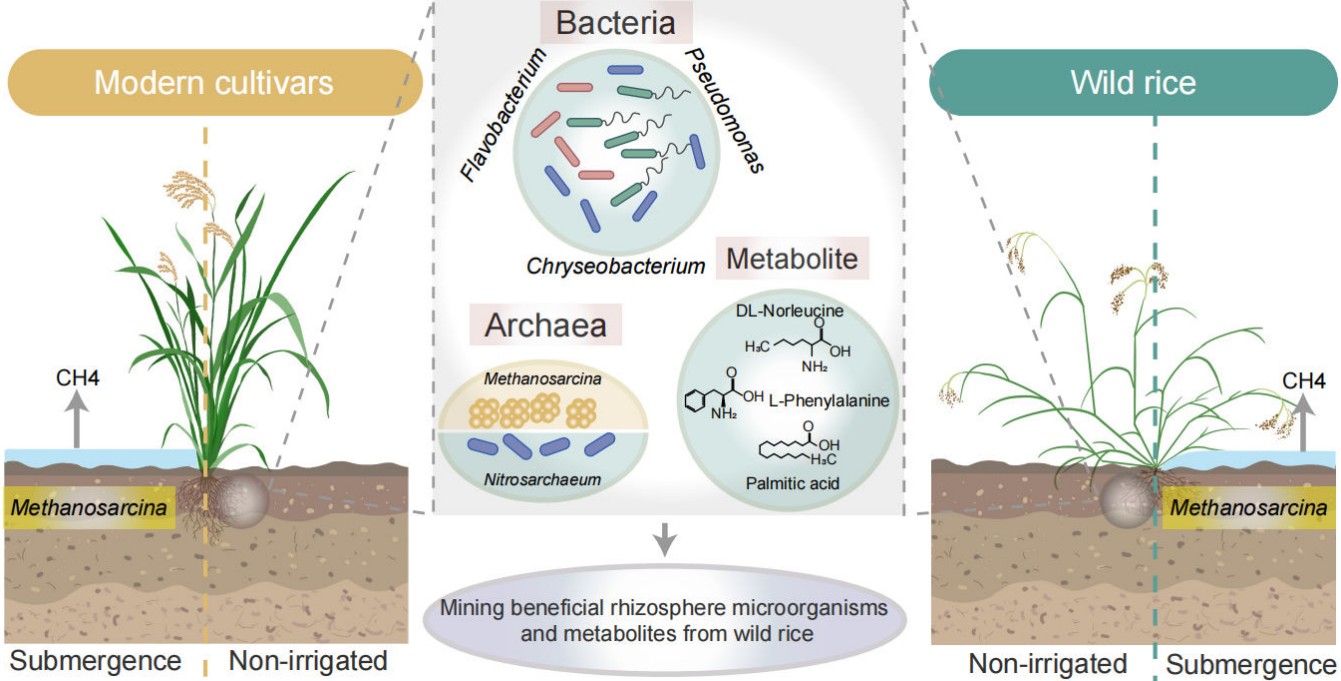

**FIG 6** The mechanistic diagram shows the candidate beneficial rhizosphere microorganisms and metabolites identified from wild rice. It highlights key microbes and metabolites in modern cultivars and wild rice under submergence and non-irrigated conditions, suggesting potential benefits for rice cultivation.

Microbial methane production accounts for approximately 74% of natural methane emissions (69). This study clarified the role of environmental conditions and host genotype in shaping microbial community structures through PERMANOVA test and multivariate statistical analysis. Host genotype had a significant impact on archaeal communities, particularly in the enrichment of *Nitrosarchaeum*. Ammonia-oxidizing archaea (AOA) are widely distributed globally and play a significant role in the global nitrogen cycle, especially in nitrification reactions (70). This suggests that wild rice may exhibit unique ecological adaptability in nitrogen cycling and resource utilization efficiency by regulating archaeal community structure. Compared to wild rice, cultivated rice was found to enrich methane-producing archaea, such as *Methanosarcina*, under both submergence and non-irrigated conditions. Compared to non-irrigated conditions, both wild rice and cultivated rice enriched more of *Methanosarcina* under irrigated conditions. Rice paddies are significant sources of methane emissions (70). Therefore, reducing methane emissions in rice fields may be of great significance. Previous research also proposed that if rice plants could be cultivated to optimize the methanogenic archaeal communities in and around the rice roots, the emissions produced in agricultural food production could be minimized (20). Our study also pointed out that we can optimize the microbiome of cultivated rice by leveraging the rhizosphere microbiome of wild rice, and reduce irrigation frequency without affecting rice growth, which can help reduce the activity of methane-producing archaea, such as *Methanosarcina*, and lower methane production and emissions. This approach has great scientific and practical value for mitigating global climate change and protecting the ecological environment.

The composition of root exudates is also strongly dependent on plant genotype. Numerous studies have shown that the total content of plant root exudates, such as soluble sugars, amino acids, and organic acids, increases with the intensity of water-limiting stress, helping the plant cope with water scarcity (11, 71). Wild rice and cultivated rice exhibited significantly different metabolic responses under non-irrigated conditions. In particular, the abundance of fatty acyls metabolites was higher in wild rice than in cultivated rice. Studies have shown that fatty acyls help plants better cope with water-limiting stress (72). Palmitic acid, a key saturated fatty acid,

participates in membrane fluidity under unfavorable environmental conditions and has been shown to improve drought resistance in wheat (73). Additionally, amino acid-based root exudates such as DL-Norleucine and L-Phenylalanine can regulate microbial communities under water-limiting stress, promoting the growth of beneficial microbes and further enhancing plant resilience. These three metabolites are primary metabolites, and primary metabolites may play a role in plant recovery (74). Wild rice may resist plant damage caused by water limitation through stronger metabolic regulation and microbial interactions in water-limiting environments.

Using metagenomic assembly methods helps uncover previously unidentified microbial species in the rhizosphere microbiome of perennial wild rice (75). We successfully obtained 214 MAGs, with 39 of them classified as high-quality draft MAGs, mainly belonging to *Pseudomonadota*, *Bacteroidota*, and *Chloroflexota*. In previous studies, *Pseudomonas* played a key role in wild rice's resistance to water limitation, belonging to the *Pseudomonadota* phylum. Among the 214 MAGs, the relative abundance of *Pseudomonadota* under non-irrigated conditions was significantly higher than in the other conditions. Thus, metagenomic assembly is an effective strategy for mining water-limiting resistance microorganisms in wild rice.

To further explore the functional potential of the rice rhizosphere microbiome, we analyzed the abundance distribution of nitrogen cycling-related genes after functional annotation using NCycDB. Previous studies have also suggested that soil moisture availability and rhizosphere effects may alter nitrogen utilization strategies in dryland soils (76, 77). Under dryland conditions, the relative abundance of the *asnB* and *nirK* genes was significantly higher in wild rice compared to cultivated rice, suggesting that wild rice may enhance nitrogen assimilation and denitrification to cope with water stress. Previous research has also shown that the abundance of the *nirK* gene is closely related to soil moisture. Under water-limiting conditions, the abundance of the *nirK* gene is three times higher than in wet soils (78). Overall, wild rice exhibited higher diversity of nitrogen cycling-related genes under dryland conditions, particularly in nitrogen assimilation and denitrification. This indicates that wild rice may have higher nitrogen utilization efficiency and environmental adaptability, particularly in water-limiting environments.

Our study reveals significant differences in the *in situ* rhizosphere microbial communities of cultivated and wild rice under contrasting water regimes; however, several limitations should be addressed. Because bulk soil is difficult to precisely define and consistently sample under field conditions, we focused on rhizosphere soil samples to maximize comparability, which, in turn, limits our ability to attribute community differences entirely to plant genotype. In addition, heterogeneity in soil physicochemical properties and agronomic practices in the field may also have influenced the observed patterns; therefore, interpretations of plant-mediated microbial selection should be approached with caution. Nonetheless, these findings reflect the characteristics of plant–microbe interactions under authentic agricultural conditions, providing direct insights for field-based crop management. Future research should incorporate bulk soil under controlled conditions, together with standardized measurements of soil physicochemical properties, to validate these associations and evaluate their applicability across a wider range of contexts.

In summary, from wild rice in natural ecosystems, we identified several key candidate microorganisms (*Pseudomonas*, *Chryseobacterium*, *Flavobacterium*) and metabolites (palmitic acid, DLnorleucine, Lphenylalanine) associated with tolerance to water stress, and we systematically compared microbial community differences between wild and cultivated rice under flooded vs nonirrigated conditions in their native environments. These candidate microbes and metabolites may have practical value for alternate wetting and drying irrigation strategies, but their functions still require experimental validation; consequently, future work will focus on elucidating the mechanisms and efficacy by which these key microorganisms modulate rice resistance to water stress.

## ACKNOWLEDGMENTS

This work was supported by the Guangxi Science and Technology Major Program (guikeAA23062085), the National Natural Science Foundation of China (U24A20369, 32100526, and 32270712), Guangxi Natural Science Foundation (2024GXNSFGA010003), and Innovation Project of Guangxi Graduate Education (YCBZ2024048).

We thank Prof. Rongbai Li for his contribution to the collection of samples.

L.-L.C. and J.-M.S. conceived and designed the study. Y.L. analyzed the data and substantively wrote the manuscript. L.-L.C., J.-M.S., Y.L., and X.X. revised the manuscript. Y.L., X.X., R.Q., R.Z., Z.-W.Z., D.-A.L., and Y.W participated in the sampling. All authors have reviewed and agreed with the paper.

## AUTHOR AFFILIATIONS

[1]State Key Laboratory for Conservation and Utilization of Subtropical Agro-bioresources, College of Life Science and Technology, Guangxi University, Nanning, China
[2]College of Agronomy and Biotechnology, Southwest University, Chongqing, China
[3]Yazhouwan National Laboratory, Sanya, China

## AUTHOR ORCIDs

Yuhong Luo http://orcid.org/0000-0001-7464-1650
Jia-Ming Song http://orcid.org/0000-0002-6636-4152
Ling-Ling Chen http://orcid.org/0000-0002-3005-526X

## AUTHOR CONTRIBUTIONS

Yuhong Luo, Data curation, Formal analysis, Investigation, Methodology, Software, Visualization, Writing – original draft, Writing – review and editing | Xiaolong Xu, Methodology, Writing – review and editing | Renfei Qiao, Methodology | Ru-Peng Zhao, Methodology | Zu-Wen Zhou, Methodology | Dong-Ao Li, Methodology | Yuhao Wen, Methodology | Jia-Ming Song, Conceptualization, Data curation, Formal analysis, Funding acquisition, Investigation, Methodology, Project administration, Resources, Supervision, Writing – review and editing | Ling-Ling Chen, Conceptualization, Data curation, Formal analysis, Funding acquisition, Investigation, Methodology, Project administration, Resources, Supervision, Writing – review and editing

## DATA AVAILABILITY

The metagenomic sequence data for this study have been deposited in Genomic Sequence Archive (GSA) (https://ngdc.cncb.ac.cn/gsa). The Illumina sequencing data are archived under the GSA accession CRA021653.

## ADDITIONAL FILES

The following material is available online.

### Supplemental Material

**Supplemental figures (Spectrum00263-25-s0001.docx).** Figures S1 to S8.
**Supplemental tables (Spectrum00263-25-s0002.xlsx).** Tables S1 to S6.

### Open Peer Review

**PEER REVIEW HISTORY (review-history.pdf).** An accounting of the reviewer comments and feedback.

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
