## [Reviewer comments · Microbiology Spectrum]

Microbiology Spectrum

Comparative analysis of rhizosphere microbiomes of cultivated and wild rice under contrasting field water regimes

Yuhong Luo, Xiaolong Xu, Renfei Qiao, Rupeng Zhao, Zu-Wen Zhou, Dong-Ao Li, Yuhao Wen, Jia-Ming Song, and Ling-Ling Chen

Corresponding Author(s): Ling-Ling Chen, Guangxi University

Review Timeline:

Submission Date:	January 25, 2025
Editorial Decision:	July 16, 2025
Revision Received:	July 26, 2025
Editorial Decision:	August 13, 2025
Revision Received:	August 15, 2025
Accepted:	August 26, 2025

Editor: Kattia Núñez-Montero

Reviewer(s): Disclosure of reviewer identity is with reference to reviewer comments included in decision letter(s). The following individuals involved in review of your submission have agreed to reveal their identity: Firouz Abbasian (Reviewer #1)

Transaction Report:

DOI: <https://doi.org/10.1128/spectrum.00263-25>

Re: Spectrum00263-25 (Exploring the role of rhizosphere microbes in wild and cultivated rice in enhancing water-limiting resistance)

Dear Dr. Ling-Ling Chen:

Thank you for the privilege of reviewing your work. Below you will find my comments, instructions from the Spectrum editorial office, and the reviewer comments.

Please note that I acted as the second reviewer as it was taking longer than usual to find an available reviewer for your submission.

Revision Guidelines

Sincerely,
Kattia Núñez-Montero
Editor
Microbiology Spectrum

Reviewer #1 (Comments for the Author):

This paper demonstrates strong technical merit. The experimental design is well-structured, and the methods employed, such as metagenomic sequencing and microbial analysis, are appropriate for addressing the research objectives. The integration of microbiome and metabolomic data provides valuable insights into how rhizosphere microbes contribute to water-limiting

resistance in both wild and cultivated rice. Overall, the technical aspects of the study are sound, and the data is analyzed using robust methodologies, making it a valuable contribution to the field.

Reviewer #2 (Comments for the Author):

This manuscript explores differences in rhizosphere microbiomes and metabolites between wild and cultivated rice under contrasting water availability conditions (non irrigated vs. submerged). The authors combine metagenomics, MAG reconstruction, and metabolomic analyses to identify key microbial taxa and functional traits potentially associated with water limiting resistance in rice.

While the topic is timely and relevant, and the overall approach is interesting, many essential methodological and contextual details are missing or insufficiently described. Also the experimental design does not fully support the conclusions. These gaps make it challenging to fully assess the validity of the results and the strength of the conclusions. I therefore recommend that the authors address the following major revisions before the manuscript can be considered for publication.

1. Lines 71-74: The authors should include data on the rice cultivation area and specify whether these regions are actually affected by water scarcity. Information about yield losses caused by this issue would also be useful to highlight the relevance of this research in a global context.
2. The introduction is largely based on general and basic information about plant-microbe interactions. The evidence presented regarding drought responses is based on plant species other than *O. sativa*. Numerous studies have assessed the microbiome and stress responses (including drought) in *O. sativa*. It is important that the authors present the state of the art specifically for rice species and clearly establish both the research problem and the novelty of their work.
3. Line 106: The scientific name should be italicized (revise across the manuscript)
4. It is important that the authors provide a more detailed description of the sampling location. They provide only one coordinate. Were the cultivation sites the same for both sampling locations? The design, based on a real cultivation system, offers a good opportunity to understand microbial dynamics, but it also introduces many variables.
5. How do the authors ensure that differences among samples are due to plant species and not site effects? The original soil microbiome may drive rhizosphere changes. For example, bulk soil samples from both sites should be included to establish the microbial baseline of each site. Additionally, validation of the observed results under controlled conditions should be suggested.
6. The methods section is unclear regarding sampling. How many samples were collected in total or per site? How was external contamination avoided? How many replicates? Were the samples collected from the same plant or from different plants? How was the cultivation time for sampling chosen? Plants recruit different microbes depending on their developmental stage. Since the authors collected samples at different times of the year, how can they ensure that the plants were at the same developmental stage (e.g., days after initial planting)? Did they confirm that the plants were healthy and growing normally before sampling?
7. Line 173: "three samples randomly selected soil sample from each condition." From how many samples were these three selected? Why did the authors choose this limited number?
8. The points mentioned above must be considered to properly evaluate the validity of the results and conclusions. The potential limitations of this study should also be discussed.
9. Line 168: The term "annotation" is not appropriate here. The method describes taxonomic affiliation. "Annotation" is generally used when genomes are assembled into MAGs and then genes are annotated.
10. Section "Data processing and statistical analysis of metabolomics": There is no description of how the metabolites were extracted, how the samples were selected, how many samples were used, or the amount of material analyzed. There is also no information on how the data were collected with LC-MS/MS (equipment, treatments, phases, columns, protocols, or standards used for internal control).
11. Section "Metagenome assembly, genome binning, MAGs classification and quantification of MAGs": It is not clear which samples were mixed. Were samples mixed before sequencing? Were reads concatenated? Which samples were mixed, and why?
12. Rarefaction curves should be provided to show whether the sequencing depth was sufficient to capture the full diversity of the samples.
13. Figure 1A-D: What do the colors of the dots represent?
14. The authors filtered the MAGs and obtained a total of 214, which is acceptable for describing genomes. However, are 214 incomplete genomes enough to perform statistical analyses comparing diversity and gene abundance? The results may be biased by the sequencing strategy or bioinformatics pipeline, favoring certain genomes that do not represent the true diversity. I recommend that the authors discuss these results with caution, as quantitative inferences cannot be made.
15. Lines 286-292: This reasoning makes sense, but soil physicochemical parameters should also be tested. Variance could be explained by soil characteristics, as these were different farms. How can this be separated from the plant genotype factor?
16. Figure S2: Diversity should be confirmed with at least two indices (e.g., Fisher, Simpson, Chao).
17. Figure S3: Presenting the data at lower and lowest taxonomic levels is important to assess variability across samples.
18. Figure S4: On which taxonomic level is this based?
19. Lines 293-301: A clear comparison with bulk soil is needed to determine whether these differences are due to plant genotype rather than sampling date or soil ecosystem. For instance, the cultivated rice field has likely been managed differently over the years, which could have changed the microbial community and altered functional redundancy in response to abiotic stress. How can we be sure these differences are related to plant genotype? This is a significant limitation of the study.
20. The authors should clearly describe how they accounted for potential confounding between the factors (plant genotype,

water condition, or their interaction), for example by applying or reporting a multifactorial statistical analysis (such as PERMANOVA with interaction terms) to distinguish genotype effects from water regime effects. In addition, the authors should explicitly state whether appropriate corrections for multiple comparisons (e.g., false discovery rate adjustment) were applied in the differential abundance analyses.

21. In general, the discussion should be improved by limiting the conclusions to those that are fully supported by the experimental design of this study.

Dear Editor and reviewers:

We greatly appreciate your consideration of our manuscript. All the comments proposed by the reviewers are very insightful and can greatly improve the quality of this manuscript. We have checked those comments very carefully and revised the manuscript accordingly. These modifications are highlighted in red in the revised manuscript. And the responses to each comment are highlighted in blue below.

We sincerely appreciate your continued consideration of our manuscript for publication in *Microbiology Spectrum* and are grateful to receive your kind decision at the earliest convenience.

----- Reviewer comments:

Reviewer #1 (Comments for the Author):

This paper demonstrates strong technical merit. The experimental design is well-structured, and the methods employed, such as metagenomic sequencing and microbial analysis, are appropriate for addressing the research objectives. The integration of microbiome and metabolomic data provides valuable insights into how rhizosphere microbes contribute to water-limiting resistance in both wild and cultivated rice. Overall, the technical aspects of the study are sound, and the data is analyzed using robust methodologies, making it a valuable contribution to the field.

Response: We appreciate your positive feedback on our experimental design and the robustness of our data analysis. Thank you for your support.

Reviewer #2 (Comments for the Author)

This manuscript explores differences in rhizosphere microbiomes and metabolites between wild and cultivated rice under contrasting water availability conditions (non-irrigated vs. submerged). The authors combine metagenomics, MAG reconstruction, and metabolomic analyses to identify key microbial taxa and functional traits potentially associated with water limiting resistance in rice.

While the topic is timely and relevant, and the overall approach is interesting, many essential methodological and contextual details are missing or insufficiently described. Also the experimental design does not fully support the conclusions. These gaps make it challenging to fully assess the validity of the results and the strength of the conclusions. I therefore recommend that the authors address the following major

revisions before the manuscript can be considered for publication.

Response: We thank the reviewer for carefully reading our manuscript, and giving us the valuable suggestions and comments.

1. Lines 71-74: The authors should include data on the rice cultivation area and specify whether these regions are actually affected by water scarcity. Information about yield losses caused by this issue would also be useful to highlight the relevance of this research in a global context.

Response: Thank you for the suggestion. We have added the rice planting area for our study region (Guangxi) and, based on the latest available data, indicated that Guangxi has indeed been affected by water scarcity. We also provided information on global rice yield losses caused by drought. The revised text (lines 72–81 in the revised manuscript) is as follows: “As the world’s largest rice producer and consumer, China contributes about 26% of global rice production (<https://www.fao.org>). Meanwhile, global rice yields are facing increasingly severe climatic challenges with the average expected loss rate due to drought can reach 13.1% (1). Against this backdrop, according to the China Statistical Yearbook 2023, rice production in Guangxi Province is likewise significant: its sown area is 1,760.8 thousand hectares and total output is 10.304 million tons, both ranking within the top ten nationwide. Notably, the disaster-affected crop area in Guangxi is 219.8 thousand hectares, of which 107.2 thousand hectares were impacted by drought, accounting for 48.8%. This highlights the potential threat that localized drought poses to yield security”.

*2. The introduction is largely based on general and basic information about plant-microbe interactions. The evidence presented regarding drought responses is based on plant species other than *O. sativa*. Numerous studies have assessed the microbiome and stress responses (including drought) in *O. sativa*. It is important that the authors present the state of the art specifically for rice species and clearly establish both the research problem and the novelty of their work.*

Response: Thanks for your comment. We have added, in lines 111–121, an overview of recent advances in the microbiome of cultivated rice (*O. sativa*) and its stress responses including drought. The revised text is as follows:

“In rice, drought stress markedly reshapes the rhizosphere microbial community structure, characterized by the systematic enrichment of Actinobacteria and Firmicutes and a concomitant decline in other taxa, while drought induces persistent shifts in the endosphere microbiome. Rice actively modulates microbial recruitment by altering root exudate metabolism including increasing abscisic acid, reducing jasmonic acid and amino acids, thereby establishing a bidirectional feedback

mechanism (2, 3). Genotype-specific responses further refine this process: drought-tolerant varieties preferentially enrich beneficial taxa through coordinated metabolic networks, whereas drought-sensitive cultivars exhibit more pronounced metabolic dysregulation. This plant–microbe co-adaptive mechanism provides a theoretical foundation for developing microbe-assisted drought-resistance strategies (2-5).”

3. *Line 106: The scientific name should be italicized (revise across the manuscript)*

Response: Thank you for the suggestion. We have italicized “*Oryza*” (lines 122) and conducted a thorough review of the entire manuscript to ensure that all scientific names are presented in italics.

4. *It is important that the authors provide a more detailed description of the sampling location. They provide only one coordinate. Were the cultivation sites the same for both sampling locations? The design, based on a real cultivation system, offers a good opportunity to understand microbial dynamics, but it also introduces many variables.*

Response: Thank you for the suggestion. The selected wild rice germplasm nursery and the cultivated rice field are both located within the same overall area of the Guangxi University campus. Because wild rice is a protected perennial Poaceae resource with growth habits that differ markedly from those of annual cultivated rice, they had to be arranged separately under real cultivation conditions. To minimize site effects, we collected cultivated rice samples from the paddy field closest to the wild rice site, with a straight-line distance of less than 1,000 m, using exactly the same sampling method as in previous studies (6, 7). The two sampling sites are spatially close, and phenological differences can be ignored; descriptions of the locations have been added to lines 168–171 in the revised manuscript. In addition, we measured soil pH and electrical conductivity at both sites during the non-irrigation period. The results showed no significant differences in soil physicochemical properties between the two sites under the same management regime and similar spatial conditions.

As the reviewer noted, a design based on real cultivation systems not only offers a valuable opportunity to reveal microbial community dynamics, but also aligns with the recent concept of “Rewilding plant microbiomes” (8). On the other hand, it inevitably introduces multiple environmental and management variables. To address this, we have added a statement of the study’s limitations in the Discussion (lines 497–505): “Due to differences in soil properties and agronomic management under real-world cultivation, microbial community structures may exhibit systematic fluctuations among plots, potentially introducing errors into analyses of community composition and function. thus, future studies should employ more rigorous bulk soil

control designs to evaluate and measurement of soil physicochemical properties and obtain more accurate candidate differential microorganisms”.

Figure R1. pH and conductivity of rhizosphere soil for wild and cultivated rice bulk soil. The data are shown as mean \pm SEM. Statistical analysis of these data was performed using the Wilcoxon test. *P* values are shown in the figure.

5. How do the authors ensure that differences among samples are due to plant species and not site effects? The original soil microbiome may drive rhizosphere changes. For example, bulk soil samples from both sites should be included to establish the microbial baseline of each site. Additionally, validation of the observed results under controlled conditions should be suggested.

Response: We thank the reviewer for carefully reading our manuscript and for the valuable suggestions and comments. Similar to the issue mentioned above, because the perennial wild rice was planted in an independent nature reserve, merely reducing the physical distance was not sufficient to completely eliminate site effects. To minimize spatial differences, we collected samples from the nearest cultivated rice field, with a straight-line distance of <1,000 m from the wild rice sampling point. As we did not simultaneously collect the corresponding soil samples at that time, this study cannot establish a microbial baseline for each site—this is a limitation shared with existing studies. As suggested by the reviewer, we have added the following to the discussion (lines 497–499): “To further verify the independent driving role of plant species on the candidate rhizosphere microbial communities, we recommend conducting validation experiments under controlled conditions to exclude interference from environmental variables.”

6. The methods section is unclear regarding sampling. How many samples were collected in total or per site? How was external contamination avoided? How many replicates? Were the samples collected from the same plant or from different plants? How was the cultivation time for sampling chosen? Plants recruit different microbes depending on their developmental stage. Since the authors collected samples at

different times of the year, how can they ensure that the plants were at the same developmental stage (e.g., days after initial planting)? Did they confirm that the plants were healthy and growing normally before sampling?

Response: We thank the reviewer for carefully reading our manuscript, and giving us the valuable suggestions and comments. We have added the following description in the revised text (lines 176–180): “A total of 12 metagenomic rhizosphere soil samples were collected across two varieties (wild rice / cultivated rice) × two conditions (non irrigated / submerged) × three biological replicates. Samples for metabolomic analysis were taken only under the non irrigated condition, totaling 12 (two varieties (wild rice / cultivated rice) × 6 biological replicates).”

To minimize exogenous contamination, all sampling followed strict aseptic procedures. Sterile gloves were worn throughout, and pre-autoclaved tools (e.g., tweezers and scissors) were used to collect soil adhering to the rice roots. Plants were carefully uprooted, loose soil was gently shaken off, and only the tightly adhering rhizosphere soil (defined as soil within 1–2 mm of the root surface) was retained. The collected soil was immediately placed into sterile 50 mL centrifuge tubes, flash-cooled on-site in a portable dry-ice container, rapidly transported to the laboratory, and stored at –80 °C until subsequent DNA extraction and sequencing (lines 180–184). All tools and containers were rigorously sterilized before use and were replaced or re-sterilized between samples to avoid cross-contamination.

To avoid confounding effects of growth stage on rhizosphere microbial communities, rhizosphere soil from cultivated rice was collected at the heading stage (lines 171). Previous studies have shown that the field rice root microbiome begins to stabilize 8–10 weeks after transplanting, i.e., during the reproductive phase (heading, grain filling, and ripening) (9). Therefore, sampling at heading effectively reduced variation attributable to developmental stage. Before sampling, field observations and records were used to ensure that all selected plants were at the heading stage (based on panicle morphology and developmental characteristics).

To confirm plants were healthy and growing normally before sampling, we checked for an upright habit with uniform tillering and firm, erect culms; dark-green, glossy, lesion- and insect-free leaves; well-developed, soil-adherent roots; and thick panicle necks with synchronous panicle development. Individuals with robust vigor and low lodging risk were classified as healthy.

7. Line 173: "three samples randomly selected soil sample from each condition." From how many samples were these three selected? Why did the authors choose this

limited number?

Response: Thank you for the suggestion. The phrase “three samples randomly selected soil sample from each condition” may be ambiguous. We intended to convey that three samples were selected for each condition. In fact, we did not randomly pick three samples from a large pool for each condition; rather, we collected a total of 12 soil samples and sequenced them according to the methods described in lines 163–184 of the manuscript. In our response to Comment 6, we have already detailed the sample size as requested. Therefore, we have revised the sentence to: “All soil DNA samples were sequenced on the DNBSEQ platform using a paired-end 150 bp (PE150) strategy (lines 189–190)”.

The choice of three biological replicates per condition was a trade-off between statistical requirements and practical resource constraints. Three biological replicates are widely accepted as the minimum in microbial community studies, allowing effective capture of biological variation and attainment of statistically significant results. Given limited funding, while ensuring statistical validity, we prioritized higher sequencing depth, obtaining an average of 17.63 Gb of sequencing data per sample. Previous studies have shown that greater sequencing depth can markedly improve the accuracy of microbial community assembly and taxonomic resolution, especially for detecting low-abundance taxa (10). By allocating our limited resources to deep sequencing of three replicate samples rather than shallow sequencing of more samples, we ensured data quality and reliability, thereby more accurately reflecting the diversity and composition of the rice rhizosphere soil microbial community.

8. The points mentioned above must be considered to properly evaluate the validity of the results and conclusions. The potential limitations of this study should also be discussed.

Response: We thank the reviewer for the valuable suggestions and comments. The limitations mentioned above by the reviewer have been added to the Discussion section.

9. Line 168: The term "annotation" is not appropriate here. The method describes taxonomic affiliation. "Annotation" is generally used when genomes are assembled into MAGs and then genes are annotated.

Response: Thank you for your insightful suggestion. We have respectfully amended “profiling annotation” to “taxonomic classification” on line 185 of the revised manuscript.

10. Section "Data processing and statistical analysis of metabolomics": There is no

description of how the metabolites were extracted, how the samples were selected, how many samples were used, or the amount of material analyzed. There is also no information on how the data were collected with LC-MS/MS (equipment, treatments, phases, columns, protocols, or standards used for internal control).

Response: We thank the reviewer for giving us the valuable suggestions and comments. We have added “Extraction and identification of metabolites” to the manuscript at lines 219–238 Details are as follows:

“Six rhizosphere soil samples each from wild and cultivated rice (biological replicates) were collected and processed as described above. d3-Leucine, 13C9-Phenylalanine, d5-Tryptophan and 13C3-Progesterone were used as an internal control for data normalization. Metabolites were separated and detected using a Waters ACQUITY UPLC I-Class Plus system (Waters, USA) coupled to a Thermo Scientific Q Exactive high-resolution mass spectrometer (Thermo Fisher Scientific, USA). A BEH C18 column (1.7 μ m, 2.1 \times 100 mm, Waters, USA) was used. In positive ion mode, the mobile phases were 0.1% formic acid in water (A) and 0.1% formic acid in methanol (B). In negative ion mode, the mobile phases were 10 mM ammonium formate in water (A) and 10 mM ammonium formate in 95% methanol (B). The gradient was: 0–1 min, 2% B; 1–9 min, 2–98% B; 9–12 min, 98% B; 12–12.1 min, 98–2% B; 12.1–15 min, 2% B. The flow rate was 0.35 mL/min, the column temperature 45 $^{\circ}$ C, and the injection volume 5 μ L. MS1 and MS2 data were acquired on the Q Exactive. The scan range was m/z 70–1050; MS1 resolution 70,000, AGC target 3×10^6 , and maximum injection time (IT) 100 ms. Data-dependent MS² was acquired on the top 3 precursor ions; MS2 resolution 17,500, AGC target 1×10^5 , and maximum injection time 50 ms; stepped NCE was 20, 40, and 60 eV. ESI source parameters were: sheath gas flow rate 40; auxiliary gas flow rate 10; spray voltage (kV) +3.80 in positive mode and –3.20 in negative mode; capillary temperature 320 $^{\circ}$ C; auxiliary gas heater temperature 350 $^{\circ}$ C” .

11. Section "Metagenome assembly, genome binning, MAGs classification and quantification of MAGs": It is not clear which samples were mixed. Were samples mixed before sequencing? Were reads concatenated? Which samples were mixed, and why?

Response: Thanks for the suggestion. The original wording could indeed be misunderstood as if the samples were physically mixed before sequencing. In fact, each sample was sequenced independently; only after obtaining the reads we performed a metagenomic co-assembly to improve contig quality. We have revised the sentence as follows: “After independently sequencing each sample, we merged the reads from all samples and performed a metagenomic co-assembly using MEGAHIT

v1.2.9 (11) with default parameters and a minimum overlap of 200 bp to obtain higher-quality contigs” (lines 265–267).

12. Rarefaction curves should be provided to show whether the sequencing depth was sufficient to capture the full diversity of the samples.

Response: We thank the reviewer for giving us the valuable suggestions and comments. We have added the rarefaction curve analysis to Fig. S2. In lines 196–197 of the manuscript, we described the method for constructing rarefaction curves as follows: “Rarefaction curves were constructed with the ‘rarecurve’ function from the Vegan package (12).” Meanwhile, we have added in lines 309–311 the following description: “The rarefaction curves reaching saturation indicate that the sequencing depth was sufficient to fully capture the rhizosphere microbial diversity in each sample.”

Figure R2. Rarefaction curve is based on the number of microbial species. Each color represents one sample.

13. Figure 1A-D: What do the colors of the dots represent?

Response: We thank the reviewer for carefully reading our manuscript, and giving us the valuable suggestions and comments. To improve clarity, we have added and revised the figure legend as follows. Colors denote the experimental groups: yellow corresponds to OS (cultivated rice under non-irrigated conditions), green to DP (wild rice under non-irrigated conditions), orange to OSW (cultivated rice under submergence conditions), and blue to DPW (wild rice under submergence conditions).

Figure R3. (A), Beta diversities of bacterial communities for all samples. Colors denote the experimental groups: yellow corresponds to OS, green to DP, orange to OSW, and blue to DPW. (D), Beta diversities of Archaea communities for all samples. Colors denote the experimental groups: yellow corresponds to OS (cultivated rice under non-irrigated conditions), green to DP (wild rice under non-irrigated conditions), orange to OSW (cultivated rice under submergence conditions), and blue to DPW (wild rice under submergence conditions).

14. The authors filtered the MAGs and obtained a total of 214, which is acceptable for describing genomes. However, are 214 incomplete genomes enough to perform statistical analyses comparing diversity and gene abundance? The results may be biased by the sequencing strategy or bioinformatics pipeline, favoring certain genomes that do not represent the true diversity. I recommend that the authors discuss these results with caution, as quantitative inferences cannot be made.

Response: We thank the reviewer for the valuable suggestions and comments. Recovery of MAGs is a powerful approach in metagenomic studies for extracting actionable insights from sequencing data (13). Our set of 214 incomplete genomes is

insufficient for robust comparisons of community-level α - and β -diversity metrics; accordingly, we did not compute those indices for the MAG dataset.

We quantified each MAG's abundance using CoverM (v0.6.1) (14)(line 276), which offers efficient coverage metrics for both contig-based (metagenomic binning) analyses. In response to the reviewer's recommendation, we discuss these results with caution. we have clarified the text at line 444 by replacing "In all rice rhizosphere soil samples" with "In all MAGs recovered from rice rhizosphere soil samples", thereby emphasizing that the subsequent conclusion pertains solely to the recovered MAG dataset rather than the entire microbial community. We have replaced the original sentence o" we found that there are many bacteria within the phylum Pseudomonadota that respond to water stress in wild rice under non-irrigated conditions" with "Within these 214 MAGs, there are many bacteria within the phylum Pseudomonadota that respond to water stress in wild rice under non-irrigated conditions (lines 447-448)".

15. Lines 286-292: This reasoning makes sense, but soil physicochemical parameters should also be tested. Variance could be explained by soil characteristics, as these were different farms. How can this be separated from the plant genotype factor?

Response: We thank the reviewer for the valuable comments. In our PERMANOVA analysis, cultivar grouping explained 44.16% of the variation in bacterial community composition, while cultivation condition explained 25.85%; the remaining 29.99% is likely attributable to other systematic variables (e.g., soil properties, growth stage, sampling location, and management practices, as noted in comments 4 and 5). The reviewer is correct that soil characteristics could account for part of this residual variance and that including additional factors would help explain community differences. However, because our original design focused on comparing different cultivars under two conditions, we did not measure rhizosphere soil physicochemical parameters at the same time. We have now acknowledged this limitation in the Discussion (lines 497 – 505 in the revised manuscript), suggesting that future studies incorporate detailed soil measurements and other environmental covariates to better partition variance in microbial community structure.

16. Figure S2: Diversity should be confirmed with at least two indices (e.g., Fisher, Simpson, Chao).

Response: We thank the reviewer for giving us valuable suggestions and comments. In accordance with the reviewer's recommendation, we have added the Chao1 and

Simpson alpha diversity indices.

Figure R4. Microbial α -diversity index (Shannon, Chao1 and Simpson) of rhizosphere soil under different conditions. “ns” represents no significant difference between groups.

17. *Figure S3: Presenting the data at lower and lowest taxonomic levels is important to assess variability across samples.*

Response: We thank the reviewer for carefully reading our manuscript, and giving us the valuable suggestions and comments. To assess variability across samples, we expanded the taxonomic levels to the species level and additionally included the genus level.

Figure R5. Relative abundance of bacterial communities in rice rhizosphere soils under different conditions at the phylum, genus, and species levels.

18. *Figure S4: On which taxonomic level is this based?*

Response: We thank the reviewer for giving us the valuable suggestions and comments. Figure S4 is at the species level, and we have added the corresponding clarification in the figure legend.

19. *Lines 293-301: A clear comparison with bulk soil is needed to determine whether these differences are due to plant genotype rather than sampling date or soil*

ecosystem. For instance, the cultivated rice field has likely been managed differently over the years, which could have changed the microbial community and altered functional redundancy in response to abiotic stress. How can we be sure these differences are related to plant genotype? This is a significant limitation of the study.

Response: We thank the reviewer for carefully reading our manuscript, and giving us the valuable suggestions and comments. Consistent with comments 4, 5, and 15, our sampling was conducted in natural ecosystems. Although we made every effort to control differences in sampling sites, growth stages, and management practices, random error could not be entirely avoided. Because re-sampling would introduce additional variables, we did not perform it. We have clarified the limitations of this experiment and provided corresponding recommendations in lines 497–505: “To further verify the independent driving role of plant species on the candidate rhizosphere microbial communities, we recommend conducting validation experiments under controlled conditions to exclude interference from environmental variables. Due to differences in soil properties and agronomic management under real-world cultivation, microbial community structures may exhibit systematic fluctuations among plots, potentially introducing errors into analyses of community composition and function; thus, future studies should employ more rigorous bulk soil control designs to evaluate and measurement of soil physicochemical properties and obtain more accurate candidate differential microorganisms”.

20. The authors should clearly describe how they accounted for potential confounding between the factors (plant genotype, water condition, or their interaction), for example by applying or reporting a multifactorial statistical analysis (such as PERMANOVA with interaction terms) to distinguish genotype effects from water regime effects. In addition, the authors should explicitly state whether appropriate corrections for multiple comparisons (e.g., false discovery rate adjustment) were applied in the differential abundance analyses.

Response: Thank you for the suggestion. We performed a PERMANOVA on the overall microbial community abundance and included the interaction between Variety and Condition in the model (Table R1). The interaction explained 13% of the variation in the microbial community.

For the differential abundance analysis, we added the following: data was analyzed using repeated-measures ANOVA, followed by Tukey’s post hoc test ($P < 0.05$). Tukey’s HSD (Honest Significant Difference) test is a multiple-comparison procedure that controls the family-wise error rate (FWER) in a single step. The corresponding methodological description has been inserted in the Methods section (lines 300–302).

Table R1. We performed a PERMANOVA on the overall microbial community abundance.

Group	Df	SumOfSqs	R ²	F	Pr(>F)
Variety	1	0.021	0.230	9.664	0.001
Condition	1	0.035	0.379	15.937	0.001
Variety:Condition	1	0.012	0.130	5.460	0.002
Residual	11	0.024	0.262		
Total	14	0.092	1.000		

21. In general, the discussion should be improved by limiting the conclusions to those that are fully supported by the experimental design of this study.

Response: We thank the reviewer for giving us the valuable suggestions and comments. We added the following limitation in lines 499–505: “Due to differences in soil properties and agronomic management under real-world cultivation, microbial community structures may exhibit systematic fluctuations among plots, potentially introducing errors into analyses of community composition and function. thus, future studies should employ more rigorous bulk soil control designs to evaluate and measurement of soil physicochemical properties and obtain more accurate candidate differential microorganisms”.

“Among the 214 MAGs (line 550)”, to avoid the ambiguity raised in Question 14, we prefixed our conclusion that the relative abundance of *Pseudomonadota* under non-irrigated conditions was significantly different from that in the other treatments.

In lines 568–577, we further note how future research can be improved: “From wild rice in natural ecosystems, we identified several key candidate microorganisms (*Pseudomonas*, *Chryseobacterium*, *Flavobacterium*) and metabolites (palmitic acid, DL-norleucine, L-phenylalanine) associated with tolerance to water stress, and we systematically compared microbial community differences between wild and cultivated rice under flooded versus non-irrigated conditions in their native environments. These candidate microbes and metabolites may have practical value for alternate wetting and drying irrigation strategies, but their functions still require experimental validation; consequently, future work will focus on elucidating the mechanisms and efficacy by which these key microorganisms modulate rice resistance to water stress”.

References

1. Guo H, Wang R, Garfin GM, Zhang A, Lin D, Liang Qo, Wang Ja. 2021. Rice drought risk assessment under climate change: Based on physical vulnerability a quantitative assessment method. *Science of The Total Environment* 751:141481.
2. Li G, Wang K, Qin Q, Li Q, Mo F, Nangia V, Liu Y. 2023. Integrated Microbiome and Metabolomic Analysis Reveal Responses of Rhizosphere Bacterial Communities and Root exudate Composition to Drought and Genotype in Rice (*Oryza sativa* L.). *Rice (N Y)* 16:19.
3. Santos-Medellín C, Liechty Z, Edwards J, Nguyen B, Huang B, Weimer BC, Sundaresan V. 2021. Prolonged drought imparts lasting compositional changes to the rice root microbiome. *Nat Plants* 7:1065-1077.
4. Tian L, Shi S, Ma L, Nasir F, Li X, Tran LP, Tian C. 2018. Co-evolutionary associations between root-associated microbiomes and root transcriptomes in wild and cultivated rice varieties. *Plant Physiology and Biochemistry* 128:134-141.
5. Pantigoso HA, Ossowicki A, Stringlis IA, Carrión VJ. 2025. Hub metabolites at the root-microbiome interface: unlocking plant drought resilience. *Trends Plant Sci* doi:10.1016/j.tplants.2025.04.007.
6. Yin Y, Wang Y-F, Cui H-L, Zhou R, Li L, Duan G-L, Zhu Y-G. 2023. Distinctive Structure and Assembly of Phyllosphere Microbial Communities between Wild and Cultivated Rice. *Microbiology Spectrum* 11:e04371-22.
7. Xu S, Tian L, Chang C, Li X, Tian C. 2019. Cultivated rice rhizomicrobiome is more sensitive to environmental shifts than that of wild rice in natural environments. *Applied Soil Ecology* 140:68-77.
8. Raaijmakers JM, Kiers ET. 2022. Rewilding plant microbiomes. *Science* 378:599-600.
9. Zhang J, Zhang N, Liu YX, Zhang X, Hu B, Qin Y, Xu H, Wang H, Guo X, Qian J, Wang W, Zhang P, Jin T, Chu C, Bai Y. 2018. Root microbiota shift in rice correlates with resident time in the field and developmental stage. *Sci China Life Sci* 61:613-621.
10. Jin H, You L, Zhao F, Li S, Ma T, Kwok LY, Xu H, Sun Z. 2022. Hybrid, ultra-deep metagenomic sequencing enables genomic and functional characterization of low-abundance species in the human gut microbiome. *Gut Microbes* 14:2021790.
11. Li D, Liu CM, Luo R, Sadakane K, Lam TW. 2015. MEGAHIT: an ultra-fast single-node solution for large and complex metagenomics assembly via succinct de Bruijn graph. *Bioinformatics* 31:1674-6.
12. Dixon P. 2003. VEGAN, a package of R functions for community ecology. *Journal of Vegetation Science* 14:927-930.
13. Zhou Y, Liu M, Yang J. 2022. Recovering metagenome-assembled genomes from shotgun metagenomic sequencing data: Methods, applications, challenges, and opportunities. *Microbiological Research* 260:127023.
14. Aroney STN, Newell RJP, Nissen JN, Camargo AP, Tyson GW, Woodcroft BJ. 2025. CoverM: read alignment statistics for metagenomics. *Bioinformatics* 41.

Re: Spectrum00263-25R1 (Exploring the role of rhizosphere microbes in wild and cultivated rice in enhancing water-limiting resistance)

Dear Dr. Ling-Ling Chen:

Thank you for the privilege of reviewing your work. Below you will find my comments, instructions from the Spectrum editorial office, and the reviewer comments.

Despite most of the concerns were addressed in the revised version of your work, some issues still need to be properly discussed.

Revision Guidelines

Sincerely,
Kattia Núñez-Montero
Editor
Microbiology Spectrum

Reviewer #2 (Comments for the Author):

The authors have addressed the majority of the previous comments, and I appreciate their effort and commitment in revising the manuscript. Substantial improvements were made to the introduction, methodological descriptions, data presentation, and clarity of sampling design, as well as the inclusion of relevant recent literature.

However, some following points remain incomplete or only partially addressed. No bulk soil samples were included for direct comparison, which limits the ability to disentangle plant species effects from site-specific environmental influences. While this omission is acknowledged in the revised discussion, no additional data were provided. Also no other potentially confounding soil properties and agronomic management variables were not quantified or controlled, besides pH and Conductance. The discussion section briefly acknowledges these methodological constraints and suggests future validation under controlled conditions. However, the potential impact of these limitations on the interpretation and generalization of the findings is not explored in sufficient depth.

My recommendation for scientific accuracy is to ensure the manuscript presents the findings in the most reliable and scientifically robust manner, with some adjustments:

1. Consider revising the title to better reflect the study's context and limitations, particularly clarifying that the patterns observed may be influenced by site-specific factors inherent to field conditions. For example: "Comparative analysis of rhizosphere microbiomes of cultivated and wild rice under contrasting field water regimes"

2. Add a short sub-section on "Limitations and Implications" in discusión where:

- o Explicitly acknowledge that the absence of bulk soil controls limits the ability to attribute observed microbiome differences exclusively to plant genotype.

- o Clarify that environmental heterogeneity in field conditions (e.g., soil properties, agronomic practices) may have contributed to the microbial patterns observed.

- o Discuss how these limitations could influence the interpretation and generalization of the results. It is important to indicate that while data suggest distinct microbial assemblages between cultivated and wild rice under contrasting water regimes, the absence of bulk soil controls and potential site-specific variability limit the extent to which these patterns can be attributed solely to plant genotype. Consequently, interpretations regarding plant-driven microbial selection should be considered with caution, and future studies under controlled conditions are warranted to confirm these associations

Dear Editor and reviewers:

We greatly appreciate your consideration of our manuscript. All the comments proposed by the reviewers are very insightful and can greatly improve the quality of this manuscript. We have checked those comments very carefully and revised the manuscript accordingly. These modifications are highlighted in red in the revised manuscript. And the responses to each comment are highlighted in blue below.

We sincerely appreciate your continued consideration of our manuscript for publication in *Microbiology Spectrum* and are grateful to receive your kind decision at the earliest convenience.

----- Reviewer comments:

Reviewer #2 (Comments for the Author):

The authors have addressed the majority of the previous comments, and I appreciate their effort and commitment in revising the manuscript. Substantial improvements were made to the introduction, methodological descriptions, data presentation, and clarity of sampling design, as well as the inclusion of relevant recent literature.

However, some following points remain incomplete or only partially addressed. No bulk soil samples were included for direct comparison, which limits the ability to disentangle plant species effects from site-specific environmental influences. While this omission is acknowledged in the revised discussion, no additional data were provided. Also no other potentially confounding soil properties and agronomic management variables were not quantified or controlled, besides pH and Conductance. The discussion section briefly acknowledges these methodological constraints and suggests future validation under controlled conditions. However, the potential impact of these limitations on the interpretation and generalization of the findings is not explored in sufficient depth.

My recommendation for scientific accuracy is to ensure the manuscript presents the findings in the most reliable and scientifically robust manner, with some adjustments:

Consider revising the title to better reflect the study's context and limitations, particularly clarifying that the patterns observed may be influenced by site-specific factors inherent to field conditions. For example: "Comparative analysis of rhizosphere microbiomes of cultivated and wild rice under contrasting field water regimes"

Response: We thank the reviewer for the valuable suggestion. To better reflect the study's context and limitations, particularly to clarify that the observed patterns may be influenced by site-specific factors inherent to field conditions, we have revised the title to the one you suggested “Comparative analysis of rhizosphere microbiomes of cultivated and wild rice under contrasting field water regimes”.

2. Add a short sub-section on "Limitations and Implications" in discusión where:

o Explicitly acknowledge that the absence of bulk soil controls limits the ability to attribute observed microbiome differences exclusively to plant genotype.

o Clarify that environmental heterogeneity in field conditions (e.g., soil properties, agronomic practices) may have contributed to the microbial patterns observed.

o Discuss how these limitations could influence the interpretation and generalization of the results. It is important to indicate that while data suggest distinct microbial assemblages between cultivated and wild rice under contrasting water regimes, the absence of bulk soil controls and potential site-specific variability limit the extent to which these patterns can be attributed solely to plant genotype. Consequently, interpretations regarding plant-driven microbial selection should be considered with caution, and future studies under controlled conditions are warranted to confirm these associations.

Response: We appreciate your insightful comments and constructive suggestions. We have added a sub-section on “Limitations and Implications” in the Discussion (Lines 573–586) in the revised manuscript.

“Our study reveals significant differences in the *in situ* rhizosphere microbial communities of cultivated and wild rice under contrasting water regimes, however, several limitations should be addressed. Because bulk soil is difficult to precisely define and consistently sample under field conditions, we focused on rhizosphere soil samples to maximize comparability, which in turn limits our ability to attribute community differences entirely to plant genotype. In addition, heterogeneity in soil physicochemical properties and agronomic practices in the field may also have influenced the observed patterns, therefore, interpretations of plant-mediated microbial selection should be approached with caution. Nonetheless, these findings reflect the characteristics of plant–microbe interactions under authentic agricultural conditions, providing direct insights for field-based crop management. Future research should incorporate bulk soil under controlled conditions, together with

standardized measurements of soil physicochemical properties, to validate these associations and evaluate their applicability across a wider range of contexts” .

Re: Spectrum00263-25R2 (Comparative analysis of rhizosphere microbiomes of cultivated and wild rice under contrasting field water regimes)

Dear Prof. Ling-Ling Chen:

Your manuscript has been accepted, and I am forwarding it to the ASM production staff for publication. Your paper will first be checked to make sure all elements meet the technical requirements. ASM staff will contact you if anything needs to be revised before copyediting and production can begin. Otherwise, you will be notified when your proofs are ready to be viewed.

Sincerely,
Kattia Núñez-Montero
Editor
Microbiology Spectrum